# Statins as Repurposed Drugs in Gynecological Cancer: A Review

**DOI:** 10.3390/ijms232213937

**Published:** 2022-11-11

**Authors:** Kai-Hung Wang, Chin-Hung Liu, Dah-Ching Ding

**Affiliations:** 1Department of Medical Research, Hualien Tzu Chi Hospital, Buddhist Tzu Chi Medical Foundation, Hualien 970, Taiwan; 2Department of Pharmacology, School of Medicine, Tzu Chi University, Hualien 970, Taiwan; 3Department of Obstetrics and Gynecology, Hualien Tzu Chi Hospital, Buddhist Tzu Chi Foundation, and Tzu Chi University, Hualien 970, Taiwan; 4Institute of Medical Sciences, College of Medicine, Tzu Chi University, Hualien 970, Taiwan

**Keywords:** statins, repurposed drugs, gynecological cancer, endometrial cancer, ovarian cancer, cervical cancer

## Abstract

Discovering new drugs is an expensive and time-consuming process, including target identification, bioavailability, pharmacokinetic (PK) tests, pharmacodynamic (PD) tests, toxicity profiles, recommended dosage test, and observation of the side effects, etc. Repurposed drugs could bypass some steps, starting from phase II trials, and shorten the processes. Statins, also known as HMG-CoA inhibitors (HMGCR), are commonly used to manage and prevent various cardiovascular diseases and have been shown to improve the morbidity and mortality of patients. In addition to the inhibitory effects on the production of cholesterol, the beneficial effects of statins on the prognosis and risk of various cancers are also shown. Statins not only inhibited cell proliferation, metastasis, and chemoresistance but affected the tumor microenvironment (TME). Thus, statins have great potential to be repurposed in oncology. Hence, we review the meta-analysis, cohort, and case-control studies of statins in gynecological cancers, and elucidate how statins regulate cell proliferation, apoptosis, tumor growth, and metastasis. Although the results in gynecological cancers remain controversial and the effects of different statins in different histotypes of gynecological cancers and TME are needed to elucidate further, statins are excellent candidates and worthy of being repurposed drugs in treating gynecological cancers.

## 1. Introduction

Gynecological cancer is any cancer that starts in a woman’s reproductive organs, including cervical, endometrial, ovarian, vaginal, and vulvar cancer. The treatments generally include surgery, radiation therapy, chemotherapy, target therapy, and immunotherapy. Combination therapy is a trend worldwide. However, discovering new drugs or targets is always the mission against cancers. There is an established setting for new drug discovery from pre-clinical results, in vitro and in vivo, to human studies, phase I and II trials, and a phase III randomized controlled trial (RCT). It is expensive and takes over 10 years in all processes [1]. Thus, if the existing drugs could be repurposed, it can dramatically reduce costs and save time, benefiting patients who suffer from these malignant and lethal diseases.

Statins, as 3-hydroxy-3-methyl-glutaryl-CoA (HMG-CoA) reductase competitive inhibitors (HMGCR), are commonly used as lipid-lowering drugs, preventing cardiovascular diseases. However, the anti-cancer properties of statins have been investigated in recent decades, showing better prognoses in various types of cancer through various mechanisms [2,3]. The evidence of the anti-cancer effects of statins in gynecological cancers is sparse and controversial, thus, we review and assess the potential of statins as repurposed drugs in gynecological cancers.

## 2. Cervical Cancer and Human Papillomavirus (HPV)

Cervical cancer is the fourth most common cancer in women and the fourth highest mortality rate worldwide. Cervical cancer diagnoses for 6.6% of all cancer types with a mortality rate of 7.5% in 2018 [4]. For diagnosis of cervical cancer from cytologic examination, the precancerous stage includes low-grade squamous intra-epithelial lesion (LSIL or mild dysplasia) and high-grade squamous intra-epithelial lesion [HSIL or moderate dysplasia, severe dysplasia, and carcinoma in situ (CIS)], and the cancer types include squamous cell carcinoma (SCC), and adenocarcinoma. The diagnosis of cervical cancer from histologic examination includes cervical intraepithelial neoplasia 1 (CIN1), CIN2, and CIN3 and cancer lesions. LSIL is relatively equal to CIN1, while HSIL is relatively equal to CIN2 and CIN3 [5].

Human papillomavirus (HPV) has been defined as a carcinogen, especially the high-risk types, and the persistence of HPV infection was a necessary etiological cause of cervical cancer [6]. High-risk HPV (HR-HPV) types include HPV16, 18, 31, 33, 35, 39, 45, 51, 52, 56, 58, 59, 66, and 68 [7]. Inoculation of HPV vaccines showed long-term efficacy and could prevent cervical cancer [8]. The ideal age for the administration of the HPV vaccines is 10 to 13 years. In low-resource settings, the simple and inexpensive way is to start with visual inspections only or in combination with HPV tests. In high-resource situations, it starts with cytologic tests (pap smear test) and HPV tests to screen cervical cancer patients [9].

## 3. Endometrial Cancer and Its Risk Factors

Endometrial cancer (EC) is the sixth most common cancer in females and the second most commonly diagnosed cancer of female reproductive organs. Around 417,000 new cases were detected, and 97,000 women died worldwide from the disease in 2020 [10] [11]. There are two main types of ECs that were characterized. Type I ECs, around ~80%, are mostly well differentiated with endometrioid histology and show a high level of estrogen receptor (ER). Type II ECs are poorly differentiated with serous or clear cell histology and show a high recurrence rate (80%~90%) within 3 years, representing a poor prognosis [12]. In addition, ECs can be low-grade (grades 1 and 2) tumors which are generally associated with a better prognosis, or high-grade carcinomas (grade 3) carrying an intermediate prognosis [13]. The risk factors for EC include high body mass index (BMI: kg/m^2^), often with other components of metabolic syndrome (e.g., hypertension, diabetes), nulliparity or infertility, early menarche, and late menopause.

The relative risk (RR) for developing EC with metabolic syndrome was 1.89 [95% confidence interval (CI) 1.34 to 2.67, *p* < 0.001] among the components of metabolic syndrome. Obesity (BMI > 30) was associated with the greatest increase in RR of 2.21 (*p* < 0.001). Other components of the metabolic syndrome linked to endometrial cancer include hypertension, with a RR of 1.81 (*p* < 0.05). [14] Type II Diabetes mellitus (DM) showed an independent risk factor for EC, with an approximate doubling of risk [Odds ratio (OR) was 2.18, 95% CI 1.40 to 3.41] [15]. Among the causes of infertility, polycystic ovarian syndrome (PCOS) showed an increase in risk (OR = 2.79, 95% CI 1.31 to 5.95) [16]. Both early menarche (RR was 2.4 for women <12 vs. ≥15 years) [17], and late menopause (RR = 1.8 for women ≥55 versus <50 years) [18] are associated with increased risk for EC.

## 4. Ovarian Cancer and Its Risk Factors

Ovarian cancer is the leading disease of death in females diagnosed with gynecological cancers. In the meantime, it is women’s fifth most frequent cause of death. There are approximately 21,750 new ovarian cancer cases in the US, comprising 1.2% of all cancer cases. The estimated number of deaths related to ovarian cancer was 13,940 in 2020 [19]. Among the ovarian cancer subtypes, type II high-grade serous carcinoma (HGSC) is the most prevalent and lethal, representing more than 70% of ovarian cancer. Type I tumor includes low-grade serous, endometrioid, clear-cell, and mucinous carcinomas, presenting at an early stage and carrying a good prognosis except for clear-cell [20]. HGSCs arise from serous tubal intraepithelial carcinoma (STIC) in the fimbriae of the tube, undergoing malignant transformation and metastasizing to the nearby ovaries and peritoneum [21,22].

The risks of ovarian cancer were increasing in postmenopausal women and those with a family history of breast or ovarian cancer. At the same time, pregnancy, lactation, and oral contraceptive pills reduced the risks [23]. Moreover, obesity was an independent prognostic factor in addition to advanced tumor staging and positive ascites cytology. The hazard ratio (HR) of overall survival (OS) was 1.871, 95% CI 1.005 to 3.486 in all ovarian cancer patients, and the HR was elevated to 2.405, 95% CI 1.335 to 4.333 in pT3 stage patients [24].

## 5. Statins, HMG-CoA Reductase Inhibitor (HMGCR) and the Role in the Tumor Microenvironment (TME)

### 5.1. Statins, Lipid-Lowering Drugs

Statins are traditionally applied in cardiovascular diseases to reduce cholesterol [25] and could be divided into two groups: type-I derivatives (from fermentation, including mevastatin, lovastatin, pravastatin, and simvastatin), and type-II drugs (from the synthetic origin, including fluvastatin, atorvastatin, cerivastatin, pitavastatin, and rosuvastatin) [26,27]. The main role of statin in the mevalonate pathway is inhibiting HMG-CoA reductase (HMGCR), resulting in the depletion of LDL cholesterol [28] (Figure 1). The statins were used between 10–80 mg, and the metabolic pathway of statins was major through CYP3A4 [29] (Table 1). However, recent studies suggested that statins could have anti-tumor effects (Figure 1 and Figure 2), from meta-analysis and bench, in vitro and in vivo.

### 5.2. Statins in Cancer

The most investigated statin in cancer is simvastatin. In general, the role of statins was tumor suppressor. Statins could induce cancer cell apoptosis through traditional caspases cascade and inhibit cell proliferation, migration, invasion, epithelial-mesenchymal transition (EMT), and chemoresistances in various types of cancer (Figure 2), including breast, lung, pancreas, and liver cancer [30]. Statins induced apoptosis of cancer cells through NFκB and the canonical caspase pathway and reduced proliferation through MEK1/2, ERK1/2, and JNK pathways [31]. Statins also induced cell cycle arrest of cancer cells by activating AMPK and increasing p21 and p27 expression [32]. Simvastatin suppresses the invasion of cancer cells by decreasing Pituitary Tumor-Transforming Gene 1 (PTTG1) [33]. Furthermore, statins could also regulate epigenetic machinery resulting in cell cycle arrest. DNMTs could be the targets of statins and the downstream p16 protein [34] and p21 [35]. In conclusion, statins showed anti-tumor progression in various cancers (Figure 2).

### 5.3. Statins in Immune Cells

Mostly, statins showed anti-inflammatory effects and enhanced the number of regulatory T cells (Treg) [36], which may result in the suppression of the Th1 immune response [37]. In addition, statin treatment reduced the Th17 population [38]. Treg obtained immunosuppressive effects on immunotherapy. However, a high dose of Atorvastatin could reduce the in vitro function of conventional T and regulatory T (Treg) cells [39]. Furthermore, statins were associated with better clinical outcomes in patients treated with PD-1 inhibitors [40]. Statins plus Th1 cytokines or dendritic cells (DC)-based immunotherapy could suppress breast tumor growth [41]. Statins could stimulate immunogenicity and promote an anti-melanoma immune response [42]. These data showed the conflicting roles of statins in immunotherapy. Thus, the roles of statins in immune cells and immunotherapy are needed to be elucidated.

### 5.4. Statins in MSCs

Statins had several effects on mesenchymal stem cells (MSCs). Statins could enhance the osteogenic differentiation, angiogenic potential, migration, homing, survival, and proliferation of MSCs [43], which may have improved therapeutic outcomes in regenerative medicine. The evidence of statins in regulating cancer-associated MSCs (CaMSCs) is limited. Simvastatin could decrease CCL3 expression from cancer cells and ICAM-1, VCAM1, IL-6, and CCL2 expression from CaMSCs, disrupting the crosstalk of the cancer cells and tumor microenvironments (TME) and inhibiting tumor progression [44]. Therefore, the roles of statins in the TME—not only in immune cells but also MSCs, especially CaMSCs— need further investigation.

## 6. Statins as Potential Anti-Cancer Agents in Gynecological Cancers

### 6.1. Meta-Analysis in EC

Statin use was associated with lower risks of EC (RR = 0.81, 95% CI 0.70 to 0.94, *p* = 0.001) but not with mortalities (HR = 0.71, 95% CI 0.64 to 0.80, *p* = 0.144) [45]. In another study, it was shown that statin use could increase overall survival (OS) (HR = 0.80, 95% CI 0.66 to 0.95) [46]. However, not all studies suggested positive results. It was shown that statin use did not reduce the risk of EC (RR = 0.88, 95% CI 0.78 to 1.00, *p* = 0.05), even in the long-term statin user (>5 years) (RR = 0.79, 95% CI 0.58 to 1.08, *p* = 0.14) [47]. There was also no protective effect on EC in another study (RR = 0.94, 95% CI 0.82 to 1.07) [48] (Table 2).

### 6.2. Cohort Studies in EC

The results of statins use in EC are controversial, including no protective effects on risks (HR = 0.67; 95% CI: 0.39–1.17) [49], OS for type I (HR = 0.92, 95% CI 0.70 to 1.2) and type II (HR = 0.92, 95% CI 0.65 to 1.29, *p* = 0.62) EC patients [50]. There was no significant association in post-diagnostic use of statins (new users) (adjusted HR 0.83, 95% CI 0.64 to 1.08) [51] and no difference between statin users and nonusers in 5-year recurrence-free survival (82% vs. 83%; *p* = 0.508), disease-specific survival (86% vs. 84%; *p* = 0.549), or overall survival (77% vs. 75%; *p* = 0.901) [52] (Table 2).

In contrast, statin use decreased the mortalities in several studies, including OS (HR = 0.41, 95% CI 0.20 to 0.82) [53], OS (HR = 0.80; 95% CI 0.74–0.88) [3], disease-specific survival (DSS) (HR = 0.63, 95% CI 0.40 to 0.99), DSS in concurrent statin and aspirin user (HR = 0.25, 95% CI 0.09 to 0.70) [54], OS in hyperlipidemic patients (HR = 0.42; 95% CI 0.20 to 0.87; *p* = 0.02), PFS (HR = 0.47; 95% CI 0.23 to 0.95; *p* = 0.04) [55], OS in continuing (pre- and postdiagnosis) users (HR = 0.70, 95% CI 0.53 to 0.92), new (postdiagnosis only) users (HR = 0.43, 95% CI 0.29 to 0.65) [56]. Furthermore, statin use decreased EC-specific mortality in type I cancers (HR = 0.87; 95% CI 0.76 to 1.00), for hydrophilic statins (HR = 0.84; 95% CI 0.68 to 1.03) and the new user (HR = 0.75; 95% CI 0.59 to 0.95) [57]. In addition, the risk of EC for statin use was decreased (HR = 0.74, 95% CI 0.59 to 0.94) [58] (Table 2).

In summary, the effects of statins in treating EC are still controversial. However, large results suggested that statins may be potent drugs to decrease the risks and mortalities of EC, and are worth performing clinical trials.

## 7. Meta-Analysis in Ovarian Cancer

Statin use was not significantly associated with the risks (RR = 0.92, 95% CI 0.85 to 1.00) but decreased the mortality (HR = 0.78, 95% CI 0.73 to 0.83) of ovarian cancer [45]. Another study showed that statin use did not reduce the risk of ovarian cancer (RR = 0.88, 95% CI 0.76 to 1.03, *p* = 0.12). Furthermore, no association was found between long-term statin use (>5 years) and the risk of ovarian cancer (RR = 0.73, 95% CI 0.51 to 1.04, *p* = 0.08) [47]. It was shown that the risks were not significantly associated with statin type (lipophilic RR = 0.88, 95% CI 0.69 to 1.12; hydrophilic RR = 1.06, 95% CI 0.72 to 1.57), histotypes of ovarian cancer (serous: RR: 0.95, 95% CI 0.69 to 1.30; clear cells: RR = 1.17, 95% CI 0.74 to 1.86), and long-term user (RR = 0.77, 95% CI 0.54 to 1.10) [59] (Table 3).

Similar to previous studies, statin use was not associated with the risk (RR = 0.88, 95% CI 0.75 to 1.03) but could significantly decrease mortality (RR = 0.76, 95% CI 0.67 to 0.86) of ovarian cancer [60]. Another study showed that statin use decreased the risks (RR = 0.79, 95% CI, 0.64 to 0.98) of ovarian cancer, especially in long-term users (>5 years) (RR = 0.48, 95% CI 0.28 to 0.80) [61]. Post-diagnostic statin use could decrease OS (HR = 0.74, 95% CI 0.63 to 0.87) and cancer-specific mortality (HR = 0.87, 95% CI 0.80 to 0.95) [62]. This could be seen in another study, showing improved OS in statin users (HR: 0.76, 95% CI: 0.68–0.85) [63]. Intriguingly, genetically proxied HMG-CoA reductase inhibition equivalent to 1-mmol/L (38.7-mg/dL) reduction in LDL cholesterol, significantly decreased the risk of ovarian cancer (OR = 0.60, 95% CI 0.43 to 0.83) as well as in BRCA1/2 mutation carriers, (HR = 0.69, 95% CI 0.51 to 0.93). [64] (Table 3).

### Cohort Studies and Case-Control Studies in Ovarian Cancer

There was no association between the risk of ovarian cancer and statin user, HR = 0.69, 95% CI 0.32–1.49 [49], OR = 0.98, 95% CI 0.87 to 1.10 [65], and HR = 0.99, 95% CI 0.78 to 1.25) [66]. Moreover, the risk was even higher (HR = 1.30, 95% CI 1.04–1.62), which was largely attributed to the effect of the hydrophilic statin, especially pravastatin (HR = 1.89, 95% CI 1.24–2.88) [58]. Statin use was not associated with mortalities of ovarian cancer, HR = 0.57, 95% CI 0.21–1.51 [67], HR = 0.90, 95% CI 0.78 to 1.04 [68], and HR = 0.90, 95% CI 0.70 to 1.15, including lipophilic statin use (HR = 0.82, 95% CI 0.61 to 1.11) and hydrophilic statins (HR = 1.04, 95% CI 0.72 to 1.49) [69], and even in the patients with hyperlipidemia (HR = 0.80, 95% CI 0.50 to 1.29) [70]. However, the mortalities were significantly decreased in non-serous-papillary subtypes (HR = 0.23, 95% CI 0.05 to 0.96) [70] (Table 3).

On the contrary, statin use decreased the mortalities of ovarian cancer, HR = 0.45, 95% CI 0.23 to 0.88 [71], HR = 0.47, 95% CI 0.26 to 0.85 [72], HR = 0.81, 95% CI 0.72 to 0.90 [73], HR = 0.74, 95% CI 0.61 to 0.91 [74], and HR = 0.66, 95% CI 0.55 to 0.81 [75], both in serous (HR = 0.69, 95% CI 0.54 to 0.87) and non-serous (HR = 0.63, 95% CI 0.44 to 0.90) histologies [75]. It was also shown that statin use decreased mortality in another study, HR = 0.76, 95% CI 0.64 to 0.89 for all patients and HR = 0.80, 95%CI 0.67 to 0.96 for patients with serous types [76] (Table 3).

Because ovarian cancer has different histotypes, statin use did not show significant differences in risks in serous and clear cell types [59], but the mortality decreased [75,76]. The results of statin use in ovarian cancer patients remained controversial. Thus, additional studies are needed to elucidate the effects of different statins on different histotypes of ovarian cancer.

## 8. Cohort Studies in Other Gynecological Cancers

The HR association between the risk of cervical cancer and statin use was 0.83, 95% CI of 0.67 to 0.99. Statin use was associated with decreased total gynecological cancer mortality, (HR = 0.70, 95% CI 0.50 to 0.98) [77]. The statin use group had a better prognosis compared with the non-user (progression-free survival: HR = 0.062, 95% CI 0.008 to 0.517; overall survival: HR = 0.098, 95% CI 0.041–0.459) in cervical cancer patients [78] (Table 4). The effects of statin use against cervical cancer and vulvar cancer are not conclusive due to too few studies and case numbers [61]. In conclusion, statin use may obtain protective effects on cervical cancer, but the evidence is too few.

## 9. The Mechanisms of the Anti-Tumor Effects of Statins on Gynecological Cancer

Simvastatin exhibits anti-metastatic and anti-tumorigenic effects in ECC-1 and Ishikawa EC cells through mitogen-activated protein kinase (MAPK) but not the Akt/mTOR pathway [79]. The drug for diabetes, metformin, combined with simvastatin, synergistically inhibited cell growth and induced Bim expression and apoptosis in RL95-2, HEC1B, and Ishikawa EC cells. The combination treatment of metformin and simvastatin upregulated phosphorylated AMPK (pAMPK) and downregulated downstream phosphorylated S6 (pS6), suggesting the mTOR pathway may regulate these anti-proliferative effects [80]. Lipophilic (lovastatin and simvastatin) but not hydrophilic (pravastatin) statins induced apoptosis in ovarian cancer cell lines A2780 and UCI 101; endometrial cancer cell line, Ishikawa; and cervical cancer cell line, HeLa, which all expressed high levels of HMG-CoA reductase [81] (Table 5).

Lovastatin and Pravastatin decreased metastasis through RhoA signaling in vitro and in vivo of SKOV3 ovarian cancer cells [82]. In addition, lovastatin and atorvastatin induced apoptosis in Hey 1B and Ovcar-3 ovarian cancer cells and suppressed anchorage-independent growth of these cells through the JNK/Rac1/Cdc42 pathway [83]. Lovastatin synergizes with doxorubicin to induce apoptosis by a novel mevalonate-independent mechanism [84]. In the mogp-TAg mice model, the promoter of oviduct glycoprotein-1 was used to drive the expression of SV40 T-antigen, and serous tubal intraepithelial carcinomas (STICs) were developed in gynecologic tissues. Lovastatin significantly reduced the development of STICs in mogp-TAg mice and inhibited ovarian tumor growth in the mouse xenograft model. Furthermore, lovastatin induced autophagy in ovarian cancer cells in vitro [85]. Simvastatin inhibited the proliferation of ovarian clear cell RMG-1 and TOV21-G in vitro and tumor growth in vivo [86]. All statins except pravastatin demonstrated single-agent activity against monolayers (IC50 = 1–35 μM) and spheroids (IC50 = 1–13 μM) of ovarian cancer cells. Furthermore, simvastatin could activate and block autophagy through the Rab7/p62/LC3-II pathway, and the lipophilic statins, simvastatin, and fluvastatin were more potent than hydrophilic statins [87]. In the ID8 syngeneic mice model, simvastatin induced apoptosis and inhibit tumor growth of ovarian cancer [88]. In a K18-gT121+/− p53fl/fl Brca1fl/fl (KpB) mouse model, a unique serous ovarian cancer mouse model specifically and somatically deletes Brca1 and p53 and inactivates the retinoblastoma (Rb) proteins; simvastatin reduced the orthotropic xenograft tumor. In vitro studies showed that simvastatin obtained anti-metastatic and anti-tumorigenic effects through MAPK and AKT/mTOR pathways [89] (Table 5).

Atorvastatin, fluvastatin, and simvastatin induced apoptosis in cervical cancer cells, CaSki, HeLa, and ViBo (established from a biopsy derived from a cervical tumor) [90]. Moreover, simvastatin reduced the phosphorylation of Raf, ERK1/2, Akt, and mTOR and prenylated Ras, resulting in the induction of apoptosis and inhibition of cervical cancer tumor growth. A combination of simvastatin and paclitaxel abolished tumor growth in vivo [91]. In addition, apoptosis and autophagy were induced by atorvastatin through the AMPK/Akt/mTOR pathway. The xenograft tumor was reduced when treated with atorvastatin [92] (Table 5).

In summary, statins showed great potential to reduce cell proliferation and tumor growth of gynecological cancer in vitro and in in vivo. Akt/mTOR is the most important pathway in regulating cell proliferation, and a combination of statins with chemo drugs could synergize the anti-tumorigenic effects. Based on this foundation, statins may be a candidate repurposed drug for gynecological cancers.

## 10. Conclusions and Perspective

The mevalonate pathway and lipid metabolism are linked to the key regulators, influencing gene expression, chromatin remodeling, cellular differentiation, stress response, and tumor microenvironment that collectively enhance tumor development [93]. Statins obtain pleiotropic roles to decrease tumor progression through mevalonate-dependent and independent pathways. Statins reduced the prenylated small GTPase and other signaling pathways, such as Akt/mTOR, to induce apoptosis and autophagy and inhibit cell proliferation and metastasis, resulting in anti-tumor development.

This review of statin use in gynecological cancers showed positive and negative results. Some studies cannot avoid confounders, including multiple comorbidities, lifestyle factors, health-related behaviors, stage and grade of disease, and other medications. The study or clinical trials of different statins (e.g., lipophilic or hydrophilic) on different histotypes of cancer (e.g., serous type or non-serous type; type I or type II) and in combination with chemo drugs are required to validate since there are only 3 trials on EC, 6 trials on ovarian cancer, compared to breast cancer which has 52 trials.

If statins are to be applied clinically to gynecological cancers, they may be used as a single agent. We advocate that using statins in combination with other drugs is more potent. In addition, the identification of the response prediction markers is just undergoing. In ovarian cancer cells, *VDAC1* and *LDLRAP1* were positively and negatively correlated with the response to statins, respectively [94].

The value of statins as therapeutic drugs against gynecological cancer is inestimable because the repurposing of inexpensive, commonly used, and FDA-approved medications to exploit their anti-cancer effects yields the development of cost-effective approaches to cancer therapy. Most important, it can directly benefit the patients, life-saving or prolonging.

## Figures and Tables

**Figure 1 ijms-23-13937-f001:**
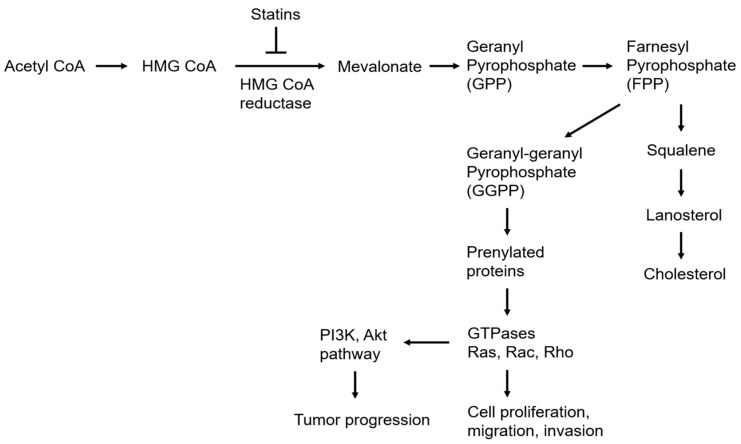
The mevalonate pathway and the role of statin in regulating tumor progression and biosynthesis of cholesterol.

**Figure 2 ijms-23-13937-f002:**
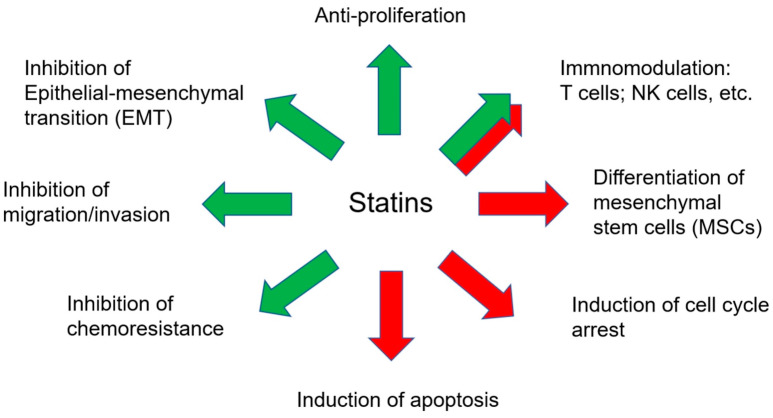
The effects of statins on anti-tumor progression and tumor microenvironment (TME). TME includes immune cells and MSCs. Green arrows represent inhibitory effects, and red arrows represent promoting effects.

**Table 1 ijms-23-13937-t001:** The doses and metabolic pathway of statins.

Drugs	Property	High Dose	Moderate Dose	Low Dose	Metabolism Pathway
Atorvastatin	lipophilic	40–80 mg	10–20 mg		CYP3A4
Fluvastatin	lipophilic		80 mg	20–40 mg	CYP2C9
Lovastatin	lipophilic		40 mg	20 mg	CYP3A4
Pravastatin	hydrophilic		40–80 mg	10–20 mg	Sulfation
Rosuvastatin	hydrophilic	20–40 mg	5–10 mg		CYP2C9
Simvastatin	lipophilic		20–40 mg	10 mg	CYP3A4

CYP3A4: cytochrome P450, subfamily IIIA, polypeptide 4. CYP2C9: cytochrome P450, subfamily IIC, polypeptide 9. High dose: reduce LDL ≥ 50%, moderate dose: reduce LDL 30–49%, low dose: reduce LDL < 30%.

**Table 2 ijms-23-13937-t002:** Clinical studies of statins in endometrial cancer.

Study Type	Findings in Statin Use Group	Results	References
Meta-analysis	Decrease risks and mortality of endometrial cancer.	Risk: RR = 0.81, 95% CI 0.70 to 0.94; OS: HR = 0.71, 95% CI 0.64 to 0.80.	[45]
Meta-analysis	Decrease mortality of endometrial cancer.	Mortality: HR = 0.80, 95% CI, 0.66 to 0.95.	[46]
Meta-analysis	No protective effect on endometrial cancer.	Risk: RR = 0.88, 95% CI 0.78 to 1.00; long-term use (>5 years) RR = 0.79, 95% CI 0.58 to 1.08.	[47]
Meta-analysis	No protective effect on endometrial cancer.	Risk: RR = 0.94, 95% CI 0.82 to 1.07.	[48]
Cohort study	No protective effect on endometrial cancer.	HR = 0.67, 95% CI: 0.39 to 1.17.	[49]
Cohort study	No protective effect on survival in endometrial cancer.	Mortality: Type I HR 0.92, 95% CI 0.70 to 1.2; type II HR = 0.92, 95%CI 0.65 to 1.29.	[50]
Cohort study	No protective effect on endometrial cancer.	Risk: HR = 0.83, 95% CI 0.64 to 1.08.	[51]
Cohort study	No protective effect on endometrial cancer.	Recurrence-free survival (82% vs. 83%); disease-specific survival (86% vs. 84%); and OS (77% vs. 75%).	[52]
Cohort study	Decrease mortality of endometrial cancer.	Mortality: HR = 0.41, 95% CI 0.20 to 0.82.	[53]
Cohort study	Decrease mortality of endometrial cancer.	Mortality: HR = 0.80, 95% CI, 0.74 to 0.88.	[3]
Cohort study	Improve DSS of endometrial cancer, especially concurrent use with aspirin.	DSS: HR 0.63, 95% CI 0.40 to 0.99; concurrent use with aspirin HR 0.25, 95% CI 0.09 to 0.70.	[54]
Cohort study	Improve OS and PFS of hyperlipidaemic high-grade endometrial cancer.	Mortality: HR = 0.42, 95% CI, 0.20 to 0.87; PFS: HR = 0.47, 95% CI 0.23 to 0.95.	[55]
Cohort study	Decrease mortality of endometrial cancer.	Mortality: continuing user HR = 0.70, 95% CI 0.53 to 0.92; new users HR = 0.43, 95% CI 0.29 to 0.65.	[56]
Cohort study	Decrease mortality of type I endometrial cancer and statin new user.	Mortality: type I HR = 0.87, 95% CI 0.76 to 1.00; hydrophilic statins HR = 0.84, 95% CI 0.68 to 1.03; new user HR = 0.75, 95% CI 0.59 to 0.95.	[57]
Cohort study	Decrease risks of endometrial cancer.	Risk: HR = 0.74, 95% CI 0.59 to 0.94.	[58]

CI: confidence interval. RR: relative risk. OR: odds ratio. OS: overall survival. HR: hazard ratio. DSS: disease-specific survival. PFS: progression-free survival.

**Table 3 ijms-23-13937-t003:** Clinical studies of statins in ovarian cancer.

Study Type	Findings in Statin Use Group	Results	References
Meta-analysis	No association with risks but decreased mortality of ovarian cancer.	Risk: RR = 0.92, 95% CI 0.85 to 1.00; OS: HR = 0.78, 95% CI 0.73 to 0.83.	[45]
Meta-analysis	No protective effect on ovarian cancer.	Risk: RR = 0.88, 95% CI 0.76 to 1.03; long-term use (>5 years) RR = 0.73, 95% CI 0.51 to 1.04.	[47]
Meta-analysis	No protective effect on ovarian cancer, regardless of the statin type, tumor histotypes: serous and clear cells, and long-term user.	Risk: lipophilic RR = 0.88, 95% CI 0.69 to 1.12; hydrophilic RR = 1.06, 95% CI 0.72 to 1.57), histotypes of cancer (serous: RR: 0.95, 95% CI 0.69 to 1.30; clear cells: RR = 1.17, 95% CI 0.74 to 1.86), and long-term user (RR = 0.77, 95% CI 0.54 to 1.10).	[59]
Meta-analysis	No association with risks but decreased mortality in ovarian cancer.	Risk: RR = 0.88, 95% CI 0.75 to 1.03; OS: RR = 0.76, 95% CI 0.67 to 0.86.	[60]
Meta-analysis	Decrease risks of ovarian cancer, especially in long-term use group.	Risk: RR = 0.79, 95% CI 0.64 to 0.98; long-term use (>5 years) RR = 0.48, 95% CI 0.28 to 0.80.	[61]
Meta-analysis	Improve OS and decrease cancer-specific mortality in ovarian cancer.	Mortality: HR = 0.74, 95%CI 0.63 to 0.87; cancer-specific mortality (HR = 0.87, 95% CI 0.80 to 0.95.	[62]
Meta-analysis	Decrease mortality of ovarian cancer.	Mortality: HR = 0.76, 95% CI: 0.68–0.85.	[63]
Meta-analysis	* Decrease ovarian cancer risks in genetically proxied HMG-CoA reductase inhibition population as well as in BRCA1/2 carrier.	Risk: OR = 0.60, 95% CI 0.43 to 0.83; BRCA1/2 carrier HR = 0.69, 95% CI 0.51 to 0.93.	[64]
Cohort study	No protective effect on ovarian cancer.	Risks: HR = 0.69, 95% CI 0.32–1.49.	[49]
Case-control study	No protective effect on ovarian cancer.	Risks: OR = 0.98, 95% CI 0.87 to 1.10.	[65]
Case-control study	No protective effect on ovarian cancer.	Risks: HR = 0.99, 95% CI 0.78 to 1.25.	[66]
Cohort study	Increase the risk of ovarian cancer, especially in pravastatin user.	Risks: HR = 1.30, 95% CI 1.04 to 1.62; pravastatin HR = 1.89, 95% CI 1.24 to 2.88.	[58]
Cohort study	No protective effect on ovarian cancer.	Mortality: HR = 0.57, 95% CI 0.21–1.51	[67]
Cohort study	No protective effect on ovarian cancer	Mortality: HR = 0.90, 95% CI 0.78 to 1.04.	[68]
Cohort study	No protective effect on ovarian cancer, neither in lipophilic nor hydrophilic statins.	Mortality: HR = 0.90, 95% CI 0.70 to 1.15; lipophilic statins HR = 0.82, 95% CI 0.61 to 1.11; hydrophilic statins HR = 1.04, 95% CI 0.72 to 1.49.	[69]
Cohort study	No protective effect on ovarian cancer with hyperlipidemia, but the mortality was decreased in non-serous-papillary subtypes.	Mortality: hyperlipidemia HR = 0.80, 95% CI 0.50 to 1.29; non-serous-papillary subtypes HR = 0.23, 95% CI 0.05 to 0.96.	[70]
Cohort study	Decrease mortality of ovarian cancer	Mortality: HR = 0.45, 95% CI 0.23 to 0.88.	[71]
Cohort study	Decrease mortality of ovarian cancer	Mortality: HR = 0.47, 95% CI 0.26 to 0.85.	[72]
Cohort study	Decrease mortality of ovarian cancer	Mortality: HR = 0.81, 95% CI 0.72 to 0.90.	[73]
Cohort study	Decrease mortality of ovarian cancer	Mortality: HR = 0.74, 95% CI 0.61 to 0.91.	[74]
Cohort study	Decreases ovarian cancer mortality, both in serous and non-serous types.	Mortality: HR = 0.66, 95% CI 0.55 to 0.81; serous type HR = 0.69, 95% CI 0.54 to 0.87; non-serous type HR = 0.63, 95% CI 0.44 to 0.90.	[75]
Cohort study	Decrease mortality in all patients and in those who were serous type.	Mortality: HR = 0.76, 95% CI 0.64 to 0.89; serous type HR = 0.80, 95%CI 0.67 to 0.96.	[76]

* Genetically proxied HMG-CoA reductase inhibition population contained single nucleotide polymorphism (SNP). CI: confidence interval. RR: relative risk. OS: overall survival. HR: hazard ratio.

**Table 4 ijms-23-13937-t004:** Clinical studies of statins in other gynecological cancer.

Study Type	Findings in Statin Use Group	Results	References
Cohort study	Decrease risks of cervical cancer; decrease mortality in total gynecological cancer.	Risk: HR = 0.83 (95% CI 0.67 to 0.99; total gynecological cancer HR = 0.70, 95% CI 0.50 to 0.98.	[77]
Cohort study	Decrease mortality of cervical cancer	Progression-free survival: HR = 0.062, 95% CI 0.008 to 0.517; overall survival: HR = 0.098, 95% CI 0.041–0.459.	[78]

CI: confidence interval. HR: hazard ratio.

**Table 5 ijms-23-13937-t005:** The preclinical studies of statin in gynecological cancers.

Treatment	Experiments	Cell Lines	Effects of Statins	Pathway/Mechanism	References
Simvastatin	in vitro	ECC-1 and Ishikawa	Anti-proliferative and anti-metastatic effects.	MAPK pathway.	[79]
Simvastatin + metformin	in vitro	RL95-2, HEC-1B, and Ishikawa	Induce apoptosis; synergized with metformin.	Bim, AMPK/S6.	[80]
Lovastatin and simvastatin	in vitro	A2780, UCI 101, Ishikawa, and HeLa	Induce apoptosis.		[81]
Lovastatin and Pravastatin	in vitro and in vivo	SKOV3	Anti-metastatic effects, reduce peritoneal dissemination.	RhoA.	[82]
Lovastatin and atorvastatin	in vitro	Hey 1B and Ovcar-3	Induce apoptosis.	JNK/Rac1/Cdc42.	[83]
Lovastatin + doxorubicin	in vitro	A2780	Induced apoptosis; synergized with doxorubicin.		[84]
Lovastatin	in vitro and in vivo	SKOV3 and OVCAR5, mogp-TAg mice	Anti-tumor growth and induce autophagy.		[85]
Simvastatin	in vitro and in vivo	RMG-1 and TOV-21G	Induce apoptosis and anti-tumor growth.	Osteopontin (OPN).	[86]
Simvastatin, atorvastatin, rosuvastatin, lovastatin, fluvastatin, pravastatin	in vitro	A2780, Igrov-1, SKOV-3, Ovcar-4, Ovcar-5 and Ovcar-8	Induce apoptosis; both activate and block the autophagy. Lipophilic statins were more potent than hydrophilic statins.	Rab7/p62/LC3-II.	[87]
simvastatin	in vitro and in vivo	SKOV3, OVCAR3, and ID8	Induce apoptosis and inhibit tumor growth.		[88]
simvastatin	in vitro and in vivo	Hey, SKOV3, and KpB mice	Anti-metastatic and anti-tumorigenic effects.	MAPK and AKT/mTOR.	[89]
Atorvastatin, fluvastatin, simvastatin	in vitro	CaSki, HeLa, and ViBo	Induce apoptosis.		[90]
simvastatin + paclitaxel	in vitro and in vivo	SiHa, C33A, HeLa, and ViBo	Induce apoptosis and inhibit tumor growth; synergized with paclitaxel.	Raf, ERK1/2, Akt, mTOR, and prenylated Ras.	[91]
Atorvastatin	in vitro and in vivo	SiHa and Caski	Induce apoptosis and autophagy and inhibit tumor growth.	AMPK, Akt/mTOR.	[92]

## Data Availability

Not applicable.

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
