# Peer review of "Statins as Repurposed Drugs in Gynecological Cancer: A Review"

_ijms, 2022, doi:10.3390/ijms232213937_

Round 1

Reviewer 1 Report

The authors present the role of statins in gynecological cancers from the viewpoint of drug repositioning. The functions are well explained using enough numbers of reports. The manuscript seems good for providing new knowledge to clinicians of gynecology. There is only one comment that doesn't need to revise. Ref. No.79 reported ECC-1 and Ishikawa cells as EC cell lines in 2014. However, in 2012, ECC-1 was revealed as a contaminated cell line. (https://pubmed.ncbi.nlm.nih.gov/22710073/) Please be careful if the paper uses the ECC-1 cell as an EC cell.

Reviewer 2 Report

Dear Authors,

This manuscript by Kai-Hung Wang and co-workers shows a review concerning the role of statins in cancer, this is a hot topic. Searching for a new application for existing drugs is more and more frequent. So this review work is sure to find a large readership. This manuscript is written in a clear, thorough, and understandable way. In my opinion, the literature search has been done precisely and exhaustively. Therefore, I suggest this manuscript be accepted for publication in its present form.

Best regards